# Preclinical and Clinical Applications of Metabolomics and Proteomics in Glioblastoma Research

**DOI:** 10.3390/ijms24010348

**Published:** 2022-12-25

**Authors:** Munazza Ahmed, Ahlam M. Semreen, Waseem El-Huneidi, Yasser Bustanji, Eman Abu-Gharbieh, Mohammad A. Y. Alqudah, Ahmed Alhusban, Mohd Shara, Ahmad Y. Abuhelwa, Nelson C. Soares, Mohammad H. Semreen, Karem H. Alzoubi

**Affiliations:** 1Department of Pharmacy Practice and Pharmacotherapeutics, College of Pharmacy, University of Sharjah, Sharjah 27272, United Arab Emirates; 2Research Institute for Medical Health Sciences, University of Sharjah, Sharjah 27272, United Arab Emirates; 3Department of Basic Medical Sciences, College of Medicine, University of Sharjah, Sharjah 27272, United Arab Emirates; 4Department of Basic and Clinical Pharmacology, College of Medicine, University of Sharjah, Sharjah 27272, United Arab Emirates; 5School of Pharmacy, The University of Jordan, Amman 11942, Jordan; 6Department of Clinical Sciences, College of Medicine, University of Sharjah, Sharjah 27272, United Arab Emirates; 7Department of Clinical Pharmacy, Faculty of Pharmacy, Jordan University of Science and Technology, Irbid 22110, Jordan; 8Department of Medicinal Chemistry, College of Pharmacy, University of Sharjah, Sharjah 27272, United Arab Emirates

**Keywords:** glioblastoma, metabolomics, proteomics, omics, clinical applications, preclinical applications, biomarkers, pharmacometabolomics

## Abstract

Glioblastoma (GB) is a primary malignancy of the central nervous system that is classified by the WHO as a grade IV astrocytoma. Despite decades of research, several aspects about the biology of GB are still unclear. Its pathogenesis and resistance mechanisms are poorly understood, and methods to optimize patient diagnosis and prognosis remain a bottle neck owing to the heterogeneity of the malignancy. The field of omics has recently gained traction, as it can aid in understanding the dynamic spatiotemporal regulatory network of enzymes and metabolites that allows cancer cells to adjust to their surroundings to promote tumor development. In combination with other omics techniques, proteomic and metabolomic investigations, which are a potent means for examining a variety of metabolic enzymes as well as intermediate metabolites, might offer crucial information in this area. Therefore, this review intends to stress the major contribution these tools have made in GB clinical and preclinical research and highlights the crucial impacts made by the integrative “omics” approach in reducing some of the therapeutic challenges associated with GB research and treatment. Thus, our study can purvey the use of these powerful tools in research by serving as a hub that particularly summarizes studies employing metabolomics and proteomics in the realm of GB diagnosis, treatment, and prognosis.

## 1. Introduction

Glioblastoma (GB) is a primary brain malignancy of the central nervous system that falls under the WHO classification of gliomas, glioneuronal tumors, and neuronal tumors and is categorized as a grade IV astrocytoma owing to its aggressive histological features [1]. GB can affect anyone, regardless of age or gender, and encompasses the primary and secondary subtypes, which differ genetically and epigenetically more so than histologically. Nonetheless, primary GB (de novo) is prevalent in older adults between 75 to 84 years, while secondary GB develops from lower-grade astrocytoma and is more common amongst younger patients [2,3]. However, the fundamental processes behind these disparities are yet to be discovered.

According to the Central Brain Tumor Registry of the United States (CBRTUS), glioblastoma was reported to be the most frequent malignant brain tumor, accounting for 14.3% of all tumors and 49.1% of those that were malignant, as assessed between 2014–2018. Furthermore, according to the incidence rate of GB by histology, GB has an incidence rate of 3.23 per 100,000 population [2]. The mainstays of contemporary GB treatment are radiation and temozolomide (TMZ); an alkylating agent, and maximal safe resection; a neurosurgical procedure aimed at resecting the tumor to the maximal extent without causing substantial neurological damage [3]. Despite this, fewer than 7% have a survival estimate of 5 years post-diagnosis, with a five-year relative survival rate of roughly 36% [2,4]. Generally, metastases are to blame for the majority of the fatalities from solid tumors [5]; however, GB almost never metastasizes. Rather, its poor prognosis has more frequently been associated with the expansion of the primary lesions, with widespread invasion into normal brain parenchyma, local pressure onto nearby areas of the brain created by mass effect, and disruption of healthy brain architecture [6].

In their recent update on the classification of tumors of the central nervous system (CNS), the WHO emphasized the role of molecular diagnostics in tumor taxonomy. Expanding on the 2016 update [7], it has added several more tumors to be defined by their molecular features, stating that in addition to the established methods of histological and immunohistochemical characterization, molecular features of CNS tumors can help analyze diagnostic alterations and aid in diagnostic accuracy [1]. This amendment is befitting as, from a molecular perspective, GB belongs to a highly heterogenic group of tumors; its histology does not necessarily correspond to the observed behavior, and interobserver variability can limit the accurate classification of CNS neoplasms [8].

### 1.1. Barriers Associated with Current Therapeutic Strategies

One of the reasons why therapeutic strategies for GB have been elusive is because tumor cells, paradoxically, depend on DNA repair mechanisms to counteract endogenous and exogenous stress, for example, from the hyperproliferating cells’ metabolic needs or that induced by chemotherapy or radiotherapy. This is a caveat since current treatment strategies such as ionizing radiation and TMZ aim at inducing DNA damage, which is frequently restored through the repair pathways such as homologous recombination and non-homologous end joining. Besides, tumor cells expressing 6- methylguanine-DNA methyltransferase (MGMT) can counteract the methylation of guanine induced by TMZ, exhibiting resistance to the drug [9]. 

Furthermore, the blood–brain barrier (BBB) must be crossed in order for novel therapeutics to effectively penetrate the tumor. In addition, GB’s biological hallmarks of pseudopalisading necrosis, angiogenesis and microvascular hyperplasia, local invasion into the healthy brain parenchyma (which renders surgical interventions difficult), and altered metabolism hamper the treatment of GB [10]. 

Hypoxia has also been proven to aid in the resistance of drugs such as TMZ and the vascular endothelial growth factor (VEGF) inhibitor, bevacizumab [11]. Furthermore, the heterogenic nature of GB cells contributes to the abundance of glioma stem-like cells (GSCs) that conform to the microenvironment posed by tumor cells—by altering their cellular phenotypes—has also been linked to hypoxia [12]. Although the precise mechanisms through which this is obtained remains unclear, generally, the resultant stress created in the microenvironment leads to the Warburg effect. The effect prevents oxidative phosphorylation and TCA cycle activity, enhancing glycolysis and lactic acid production [13]. Despite the lack of understanding of the mechanisms contributing to the influence of hypoxia on the metabolomic reprogramming of tumor cells, research is adept at accurately modelling and representing the metabolomics of the tumor microenvironment [14].

Despite such understanding, to increase the likelihood of managing cancer, novel therapeutic approaches are still desperately needed. Although histopathological assays are regarded as the gold standard in the detection of cancer, these tests give subjective results, highlighting the need for more precise procedures that can generate reliable and objective data [15]. Considering this, for the purpose of detecting genes, mRNA, proteins, and metabolites, omics analytical techniques such as genomics, transcriptomics, proteomics, and metabolomics have been developed. 

### 1.2. Metabolomics and Proteomics

Omics studies, including genomics, epigenomics, transcriptomics, proteomics, and metabolomics, have proven quick advancement in the early diagnosis of diseases. Each type of multi-omics data presents a series of differences that can be utilized as markers of an underlying disease process and also give an observational view of which biological pathways are affected when the disease and control groups are compared [16,17,18]. On the other hand, the dependence on one type will give limited information about the cause of these changes, while a combination of omics technologies will help in addressing the causative changes that lead to underlying disease [19].

Metabolites reflect the true condition of the cells and tissues as they mirror the underlying metabolic processes incorporating both the genetic (internal) and environmental (external) variables. Metabolomics comprehensively analyzes a myriad of metabolites in a cell, tissue, or biological fluid, capturing the metabolic composition at a given point of the physiological status of the cell—i.e., the metabolome [20]. In this context, cancer therapy can be tailored to each patient’s unique responses using metabolite profiling. The relationship between the metabolic profile, genetics, and phenotype makes this conceivable [21]. Proteomics explicates the protein products and their posttranslational modifications, and helps document the proteome’s spatiotemporal organization as well as localization and interaction with protein products [22]. However, it is important to note that the field of “omics” is not mutually exclusive and has extensively been used in research together to present a more comprehensive outlook into disease therapeutics. Metabolomics and proteomics essentially aim at correlating the altered metabolites or proteins to the underlying pathophysiological abnormalities. In this context, Marziali et al. reported that, in a subset of patients, the metabolic characteristics of GB cells, as determined by NMR analysis, provided insight into the tumor’s genomic/proteomic landscape and, ultimately, into the subtype of GB and therefore the patient’s clinical fate [23].

Therefore, by highlighting studies employing omics technology to assist in the detection and analysis of eminent GB biomarkers in clinical and preclinical studies, our review aims to emphasize the significant impact this tool has brought forth into scientific research. In addition, it signifies the important contributions of the “omics” integrative approach to alleviate some of the therapeutic hurdles of GB treatment and research by determining the molecular mechanisms of the disease via identifying the biologically significant patterns in a multitude of pathways involved in GB. Thereby, this review can furnish future GB studies and ultimately encourage the utilization of this functional technology to pursue further into optimizing GB diagnosis, prognosis, therapeutics, and treatment outcomes.

## 2. Metabolomics and Proteomics Workflow

Certain variations between the methodology of metabolomics and proteomics exist, despite the fact that both often share an overarching methodology. Figure 1 illustrates the schematic workflow for metabolomics and proteomics. However, depending on the purpose of a metabolomics investigation, untargeted and targeted metabolomics can be utilized [24].

Non-targeted studies enable a more thorough analysis of metabolomic profiles, while targeted studies restrict research on a selected number of known compounds. Although a small proportion of metabolites are identified using the targeted research, current techniques, nonetheless, enable the construction of comprehensive metabolic profiles encompassing hundreds of compounds. However, far more compounds are examined in non-targeted investigations because of the necessity to process large datasets that contain hundreds of metabolic signals, due to the fact that only a small subset of these metabolic signals ultimately serve as candidate biomarkers [25].

### 2.1. Targeted Metabolomics

Where the metabolomes of two groups (control and treatment) are identified and compared to determine the changes in metabolite profiles that may be important to specific biological circumstances, targeted metabolomics is the employed method [26]. Contrary to untargeted metabolomics, the metabolites in the targeted technique are chemically defined and biochemically annotated at the beginning of the investigation. However, it is only in the presence of an actual chemical standard for the metabolite that this technique can successfully be carried out. For the purpose of creating calibration curves for each of the metabolites under investigation, the quantification of metabolites is carried out using chemical standards. Here, sample preparation employs techniques that can be tailored to retain relevant metabolites while excluding irrelevant biological species and analytical artifacts [25].

After isolating metabolites of interest from a biological specimen to prepare the sample, an appropriate analytical platform is employed to detect, identify, and characterize utilizing databases such as METLIN, the human metabolome database (HMDB), and MetaboAnalyst, with the most common platforms being mass spectrometry (MS) coupled with liquid, gas chromatographic techniques, and nuclear magnetic resonance spectroscopy (NMR). Table 1 lists some of the analytical platforms used in metabolomics. Following that, statistical techniques are employed, such as the principal component analysis (PCA), to identify and quantify possible biomarkers, which are then confirmed for their relevance to a certain biological function or metabolic pathway [24]. 

### 2.2. Untargeted Metabolomics

Untargeted metabolomics is a broad method that uses extensive metabolite profiling to define as many metabolites as possible from samples without regard for bias. It is usually considered when the aim is to detect unpredictable changes in metabolite concentrations. To increase the possibility of detecting these unforeseen changes, it is preferred that the maximum number of metabolites be detected, as much as is feasible. A single analytical technique, however, would be unable to identify every metabolite present in a biological system. Thus, to increase the number of metabolites discovered and enhance metabolome coverage, it is consequently important to integrate multiple analytical procedures (such as complementary HPLC methods) [25].

Therefore, untargeted metabolomics can generally be split into three steps: metabolite profiling, identification, and functional interpretation. Sample preparation precedes the identification of the metabolites with statistically significant differences. It entails the extraction of the biological sample’s metabolites in a suitable solvent for analytical analysis. The extracted sample is then examined using the proper analytical technique (for example, LC–MS). The output of the mass spectrometry analysis is a chromatogram, and the statistical analysis uses the peak area of each metabolite as a parameter to characterize the concentration differences between the various biological samples that were analyzed [25]. Compound identification entails identifying and annotating identified compounds in the profiling stage by scanning the metabolomic databases previously mentioned. Since there is no comparison with calibration curves created from chemical standards, this method is known as relative quantification. Utilizing calibration curves is essential, however, for complete quantification. Identification of metabolites is followed by interpretation, which seeks to correlate the significant metabolites with critical pathophysiological processes or pathways [24]. Any potential biomarker found through metabolomic investigations is further validated by clinical trials or field studies [27]. 

**Table 1 ijms-24-00348-t001:** Metabolomics analytical platforms.

Technique	Characteristics	Advantages	Disadvantages	References
NMR	Nuclear magnetic resonance (NMR). 1D/2D ^1^H NMR usually employed in metabolomics studies.	Brief analysis time. Absolute metabolite quantification. High reproducibility. Intrinsically a quantitative technique since the signal strength is directly correlated with metabolite concentrations.	Its relatively low sensitivity can miss low-abundance metabolites. Signal overlapping due to the lack of a prior separation system. Typically, nonselective analysis is done with NMR. Major complications are presented by peak overlaps from numerous measured metabolites.	[28,29]
GC-MS	Gas chromatography–mass spectrometry (GC-MS). Volatile metabolites or those rendered volatile can be analyzed by GC-MS.	Relatively cheap. Better stability and separation efficiency. Being able to monitor highly hydrophobic and volatile compounds not ionized in the ESI source of LCMS/MS.	Unable to directly analyze nonvolatile, polar, or thermally labile drugs. Many biological substances are either too big or too polar to be examined using this method.	[30,31,32,33]
LC–MS/MS	liquid chromatography–mass spectrometry (LC–MS). High sensitivity. Biofluids like urine can be introduced directly into the LC system. Multidimensional can study the metabolome and lipidome in the same run.	Multidimensional—can study the metabolome and lipidome in the same run. High sensitivity. Biofluids like urine can be introduced directly into the LC system.	Greater operational costs. Lower concentration sensitivity. More restricted sample throughput.	[34]
LC–MS–NMR	Liquid chromatography-(LC)–mass spectrometry (MS)–nuclear magnetic resonance (NMR) Good spectral resolution and excellent metabolite identification ability.	Good spectral resolution. Excellent metabolite identification ability.	Different rates of exchange with deuterium are possible for analytes having exchangeable or “active” hydrogens since NMR uses deuterated solvents. The analyst should be aware of this possibility since it might cause a number of clustered molecular ions. The NMR component has high sample mass requirements.	[35,36,37]

### 2.3. Proteomics Workflow

The majority of proteomics research is conducted using liquid chromatography with mass spectrometry (LC–MS), which combines direct sampling and proteomic analysis using mass spectrometry imaging (MSI), and has gained popularity since it allows for live single-cell analysis with extremely low sample needs and delivers great sensitivity [38]. 

Liquid chromatography–mass spectrometry (LC–MS) is further divided into top-down and bottom-up methods. The bottom-up technique is the most widely utilized and is the method of choice for identifying unknown proteins; also known as “shotgun proteomics.” Proteins are cleaved into peptides by enzymatic digestion in the bottom-up technique following sample extraction and processing. This is followed by multidimensional LC (MDLC) separation and characterization by tandem MS (MS/MS), as well as protein identification utilizing statistical databases [39].

A top-down approach, on the other hand, intends to first separate protein mixtures and then sequence the intact proteins. Individual post-translational alterations may be identified and quantified using this technique, owing to the lack of the enzymatic digestion step in top-down proteomics [40]. Here, the electron spray ionization (ESI) technique is generally preferable as it generates multiple charged precursor ions for a more effective dissociation of bulky protein ions, and gives more MS/MS possibilities than matrix-assisted laser desorption/ionization (MALDI), which primarily generates single charged species [41]. 

## 3. Clinical and Preclinical Applications of Metabolomics and Proteomics in Glioblastoma Research

When compared to the other “omic” counterparts, metabolomics and proteomics have an upper hand, simply because the metabolome and proteomes are more “proximal” to the disease in question [42]. Furthermore, the ability to modify proteins and metabolites makes them suitable therapeutic targets, which is perhaps the most crucial factor ascribed to the barriers associated with GB therapeutics.

Although genomic analyses can predict mutations that can be targeted, clinically monitor therapeutic efficacy and outcomes, and describe treatment resistance mechanisms, advanced diseases such as cancer render the identification of a definite relationship with genetic variants difficult [43]. Genomics and transcriptomics delve into gene expression to provide certain “patterns” that can be assigned to specific diseases, and it is understood that the assessment of genes alone is inadequate to understand disease development in detail, particularly for heterogenic diseases such as GB. Hence, the study of proteins, pathways, metabolic networks, and metabolism has received more attention recently.

### 3.1. Metabolomics to Elucidate the Molecular Mechanisms

Medium-chain fatty acids (MCFA) have demonstrated positive effects on a variety of brain conditions, including traumatic brain injury [44], Alzheimer’s disease [45], diabetes [46], and cancer [47], when included in ketogenic diets. Despite the interest in its effects, little is known about how MCFAs are metabolized in the brain. Therefore, the metabolic changes of U87MG glioma cells for 24 h using a gas chromatography–mass spectrometry (GC–MS)-based metabolomics technique, in conjunction with multivariate data analytics was conducted by Damiano et al., after the addition of octanoic (C8) and decanoic (C10) acids; the major components of the ketogenic diet [48]. The investigation revealed notable variations in the metabolism of U87MG cells following the addition of the fatty acids and identified a number of metabolites whose concentrations varied between the two treated cell groups. The findings showed that, whereas C8 affected mitochondrial metabolism, increasing the generation of ketone bodies, C10 mostly affected cytosolic pathways by promoting fatty acid synthesis. C10 has been shown to be a PPARγ agonist, and octanoic acid, is unlikely to share this property with decanoic acid. It could be assumed that C10 increases de novo lipogenesis in U87MG cells via a PPARγ-mediated mechanism [49], given that PPARγ is able to control the lipogenic transcription factor SREBP1c, which in turn induces the production of the lipogenic enzyme fatty acid synthase. These findings would be of significance given that fatty acid synthesis, which serves to create fundamental lipid molecules for membrane formation, might support cell proliferation. Fatty acids, on the other hand, can kill cancer cells because of effects akin to those of detergents [50].

Likewise, the underlying mechanisms of natural anti-cancer agents were explored in another study by identifying the prospective markers and exploring their targets via employing ^1^H NMR-based untargeted metabolomics approach, coupled with real-time quantitative reverse transcription polymerase chain reaction (qRT-PCR) and flow cytometry [51]. Influenced by the chemo-sensitization ability of natural polysaccharides [52,53,54,55,56,57], Cibotium barometz (CB), a tropical plant rich in polysaccharides was tested for the effect of heat processing on the plants’ chemosensitization ability. Differences in the mechanisms of both the processed (PCB) and raw (RCB) batches were uncovered, where the former was able to significantly enhance the sensitivity of U-87 cell lines to TMZ, exhibit stronger toxicity to U-87 cells with TMZ, cause cell cycle arrest, spark metabolic changes, deplete GSH; leading to reactive oxygen species (ROS) accumulation, and cell death [51].

### 3.2. Metabolomics to Unravel the Resistance Mechanisms

Since MGMT counteracts the effect of TMZ and is accountable for imparting resistance [58], the metabolic alterations and differences in metabolite concentrations brought upon by the treatment of GB with TMZ and the MGMT inhibitor, lomeguatrib, were studied. The data obtained added to the diagnostic and therapeutic understanding of TMZ resistance in GBs. Furthermore, the analyses of TMZ-resistant and sensitive GB cell lines provided new insights on the mode of action of both drugs [59]. 

Another factor behind TMZ resistance has been attributed to the specificity protein 1 (Sp1), which reportedly encourages the development of drug resistance in GB by regulating cell senescence, DNA damage response, and tumor angiogenesis, among others [60,61]. Since this is done by catalyzing the metabolism of cholesterol to neurosteroids, a study set out to further elucidate the function of Sp1 in regulating arachidonate metabolism [62]. By performing MS and RNA-sequencing to analyze metabolite levels and gene expressions, researchers were able to conclude that arachidonic acid (AA) metabolism regulated by Sp1 participates in TMZ resistance via the cyclooxygenase (COX) pathway. It was further revealed through targeted arachidonate metabolome analyses that prostaglandin E2 (PGE2) is pivotal for GB recurrence and TMZ resistance. This can be explained by the fact that AA metabolism is crucial in the synthesis of PGE2, which is upregulated in multiple types of cancer [63]. Additionally, the generation of bioenergy inside mitochondria is leveraged by fatty acid β-oxidation (FAO); a primary step in lipid metabolism and a means by which tumors develop in nutrient-impoverished environments. Through untargeted metabolome analysis, increased FAO-associated metabolites were also observed in TMZ-resistant GB cells [64].

Regrettably, despite rigorous treatment methods, GB recurrence is unavoidable, and the only treatment that significantly extends patient survival is post-surgical radiation therapy (RT), which is also liable to resistance (39). Thus, identifying the underlying causes of RT resistance may improve the prognosis for GB patients. Bailleul et al. addressed this and concluded that radiation-induced metabolic reprogramming of GB cells involves the de novo serine synthesis pathway (SSP) [65]. This was done by performing high-performance liquid chromatography–mass spectrometry (HPLC/MS)-based metabolomics assays in patient-derived GB samples to investigate the changes in metabolite levels induced by radiation. Targeted metabolomics coupled with 13C-labeled glucose was adopted to track glucose toward antioxidant species and SSP intermediates. After radiation, gene expression levels were assessed using molecular biology techniques. Through the use of TCGA pan-cancer analysis, the researchers examined copy number alterations (CNA) and amplification of SSP enzymes in low-grade gliomas and GB. 

When the GB tumors were assessed, the majority (82%) had the phosphoserine phosphatase (PSPH) gene amplified or copy number alterations (CNA). The former encodes for a key enzyme in serine synthesis [66], while the latter includes the deletion or amplification of genomic segments, which are highly prevalent in cancer and contribute significantly to the development and progression of the disease [67]. Radiation raises serine, cystine, and glycine levels in gliomaspheres, which are accompanied by reduced levels of glutathione (GSH). Moreover, nucleotide and precursor levels in radiation-treated GB cells are elevated downstream of the SSP; possibly supporting DNA damage repair pathways following radiation.

### 3.3. Metabolomics and GB Characterization

Metabolic reprogramming is not only a means of biomass generation and energy supply through the accretion of metabolic intermediates, but it also serves as a controller for the preservation of the oncogenic state of cancer cells [68]. Since most tumors have metabolic heterogeneity emanating from GSCs—which have also been reported as facilitators of therapeutic resistance, neurosphere assays (NSA) are carried out in vitro to study such cells under defined culture conditions [69,70]. 

Upon analyzing the transcriptome and metabolome of cell lines U87 and NCH644 exposed under monolayer and neurosphere culture conditions, a study was able to identify important metabolic pathways and genetic signatures related to GSCs, concluding that neurosphere and monolayer cells considerably vary in their gene regulation and metabolism [71]. Furthermore, in neurosphere conditions, the expression of differentiation-associated genetic signatures were found to be repressed. The two cell lines used in the study displayed deregulation of the arginine biosynthesis. In NS conditions, this was noted to be a crucial metabolic pathway since it facilitates and maintains the stem-like phenotype. Nonetheless, diverse metabolic outcomes were noted between U87 and NCH644, possibly related to the distinct metabolic needs of each cell line. The study highlights the possibility of developing metabolism-targeting therapies and that modified metabolic processes modulate cancer proliferation and cell survival. 

Evidently, so far, metabolomics is not usually employed independently in studies. An integrative methodological approach is usually adopted that includes, but is not limited to, genomics and proteomics. This aids in the meaningful analysis and interpretation of data needed, which is crucial to yield meaningful results. 

For instance, 99 treatment-naïve GBs were collected from Clinical Proteomic Tumor Analysis Consortium (CPTAC) to be prospectively investigated by integrating metabolomic and proteogenomic data from different platforms such as RNA-seq, lipidome, metabolome, and proteome. MS was utilized to quantify protein, acetylation, and phosphorylation, whereas label-free gas and LC–MS was used to measure metabolome and lipidome levels. Integrating metabolomic and proteomic data revealed different global metabolic alterations and diverse lipid distributions across subtypes in IDH-mutated cancers. Moreover, through the integrative approach, receptor tyrosine kinase (RTK)-altered tumors were revealed to have: phosphorylated phospholipase C gamma 1 (PLCG1) and protein tyrosine phosphatase non-receptor type 1 (PTPN1), while classical GBs had low macrophage composition with enhanced histone (H2B) acetylation, stromal cells lacked the mesenchymal subtype EMT signature, and finally that there are four immune subtypes of GB, each with a different population of immune cells [72]. 

In a similar effort, three glioma cell lines, each with distinct stemness, were investigated for their metabolic signatures to identify the molecular mechanisms fundamental to the differentiation property of GSCs [73]. Using LC–MS based targeted and untargeted metabolomics approach, the differentially expressed metabolites of U87MG stem-like cells (SLCs) relative to U87MG stem-like cell differentiation cells and U87 glioma cells were characterized. The tricarboxylic acid cycle and oxidative phosphorylation, as well as the altered glycerophospholipid, nucleotide, glutathione, carnitine, and tryptophan metabolisms, all contribute to the energy metabolic pathways. The study raises the possibility of employing cell metabolomics to clarify the biochemical mechanisms of the detrimental property of GSCs [73]. 

### 3.4. Metabolic Phenotyping in Diagnostics

Given that metabolomics unravels the biochemical activities of a biological system with the utmost sensitivity and spatial precision, obtaining metabolomic profiles from biospecimens like tissue samples, blood, urine, or cerebrospinal fluid might be used to explain a disease’s associated metabolic phenotype [74,75,76].

With the aim of highlighting the GB molecular landscape, Gillard et al. performed a comprehensive metabolomic analysis by utilizing both plasma and tissue samples from GB patients [77]. The variations in the metabolic profiles of GB and control samples were investigated using three distinct metabolomics techniques. On the tissue samples, untargeted mass spectrometry imaging, and on the plasma samples, both untargeted and targeted analysis. Four classes of metabolites were identified as representatives of metabolic remodeling in tissue and plasma samples from GB patients, including phospholipids, acylcarnitines, triacylglycerols, and sphingomyelins. The study was also able to investigate both local and systemic metabolic phenotypes and reveal metabolic patterns that could aid in the diagnosis of early non-invasive GB [77]. 

Similarly, efforts to identify key biomarkers, regardless of their novelty, that can aid in diagnostics have been the recent focus of most GB research. Simply put, studies are pursuing opportunities to seek diagnostic biomarkers from previously established metabolome data. A recent example of this is a study where researchers postulate that through microdialysis, IDH1-mutant tumors with a distinct metabolomic profile may reveal a metabolomics-based biomarker response to TMZ [78]. LC–MS was used to perform an untargeted metabolomic profiling analysis on perfusates obtained from intracranial tumors in mice. Compared to the control in the experiment, tumor-specific metabolites were noted, in addition to high levels of metabolites such as dimethylarginine, a multitude of amino acids, and 5-methylthioadenosine. Interestingly, upon TMZ treatment, histidine and creatine metabolism changed. Through this study, researchers hope that prolonged in vivo intratumoral microdialysis might produce pharmacodynamic metabolic biomarkers, valuable in therapies for IDH-mutant gliomas [78]. 

### 3.5. Pharmacometabolomics 

Metabolic phenotyping is generally referred to experiments that employ an intervention followed by an interpretation of the resultant metabolic profiles, i.e., in a diagnostic manner. On the contrary, the study of metabolite profiles preceding an intervention, when used to predict the results of the forthcoming intervention, is termed predictive metabolic phenotyping. Pharmacometabolomics is the term coined for experiments where the said intervention is drug administration [15,16]. Justifiably, pharmacometabolomics is rapidly gaining prominence in the realms of drug discovery and personalized medicine.

For example, Dastmalchi et al. performed NMR analysis on urine samples collected from GL261-gp100 neoplastic mice that were then given an anti-programmed death 1 (anti-PD-1) or bone-marrow-derived dendritic cell (DC) vaccine. Upon sparse partial least squares discriminant analysis (sPLS-DA), mass metabolic changes were observed to be induced with immunotherapy. DCs play a critical role in triggering the immune system, thereby providing an excellent tool for enhancing endogenous anti-tumor responses, which can ultimately aid in the efficient removal of malignancies [79]. Moreover, anti-PD-1 antibodies are a group of immune checkpoint inhibitors that interfere by blocking the programmed death 1 receptor on the surface of lymphocytes, eventually increasing the activity of these immune cells [80]. The study was able to identify the principal drivers of the differences in the metabolic changes, suggesting the possibility of identifying individuals who will respond to immunotherapy for GB and eventually portray better treatment outcomes [81]. 

In another study, Cuperlovic-Culf et al. looked into how two different GB cell lines’ metabolisms were affected by HDAC inhibitors and inhibitors of Silent mating-type Information regulator proteins (SIRT)—a NAD+ dependent deacetylase [82]. Epigenetic regulation of gene expression through lysine acetylation is catalyzed by histone-modifying enzymes (HME) [83]. This reversible acetylation is a crucial process that is involved in the regulation of the chromatin structure, a lack of which would cause atypical acetylation levels [84]. Such atypical levels can be precipitated through histone deacetylases (HDAC), which, in contrast to HMEs, deregulate the epigenetic mechanisms and as a result, cause malignancies such as GB. Accordingly, HDAC inhibitors predominantly exhibit antiproliferative activity [85]. 

Proton nuclear magnetic resonance (^1^H NMR) analysis and principal component analysis (PCA) were performed for the metabolites generated by both the treated and control samples of U373 and LN229 cell lines. The inhibitors showed intriguing effects on glycolysis, mitochondrial metabolism, and fatty acid synthesis, implying that protein deacetylases may play a role in metabolism regulation. However, distinct changes in the metabolic profiles upon treatment with protein deacetylators were seen between the GB cell lines used, which can be utilized for developing biomarkers for treatment planning in the near future. Moreover, the study was able to explore the metabolic indicators of the deacetylase therapy on the effectiveness of such treatments and suggest that the addition of HDAC and SIRT treatments to TMZ therapy could be used for combined treatment [82]. Additional pharmacometabolomics studies are summarized in Table 2.

### 3.6. Proteomics to Elucidate the Molecular Mechanisms

Although the literature on genetic alterations is well established, information on proteomic alterations of gliomas of different subclasses is scarce. Buser et al. used quantitative proteomics by MS to examine glioma samples of various grades obtained from patients and discovered a coordinated decrease of numerous components of the endocytic machinery [99]. Furthermore, downregulation of endocytosis was frequently observed in the additional subtypes observed in the study. Endocytosis is the process by which external cargo is taken up by carriers or membrane-encapsulated transport vesicles. The cargo is either internalized by clathrin-independent or clathrin-mediated endocytosis (CIE or CME) depending on certain characteristics [100]. According to the proteomic findings, major components of the endocytic machinery involved in the CIE and CME pathway were significantly downregulated. This is known to impair efficient receptor internalization, creating a vantage point for glioma to progress due to the extended RTK signaling from the cell surface [100]. Downregulating the several proteins involved in CIE and CME could be a mechanism for GB to competently limit RTK downregulation. Moreover, the data repository generated from the proteomics performed could be beneficial in discovering and implementing innovative treatment options for GB [99]. 

Utilizing the isobaric tagging for relative and absolute quantification (iTRAQ) based- MS, proteomic analysis of grades II, III and IV glioma samples from patients was conducted by Gollapalli et al. [101]. A comprehensive proteomic analysis exposed significantly altered proteins that, upon bioinformatics analysis, revealed various perturbed metabolic pathways, including lactate metabolism, glycolysis, blood coagulation pathways, and TCA-cycle. Moreover, proteins involved in antiapoptotic and redox reactions, to name a few, were upregulated in gliomas. Upon the comparison of the proteomes of different grades of glioma, proteins involved in several metabolic pathways were significantly altered. The metabolic pathways identified were linked to neurodegenerative disorders, amino acid, carbohydrate, and pyruvate pathways, to list a few. Other proteins discovered to be changed in gliomas were involved in cell cycle control and proliferation. Through proteomics, several proteins seen in different grades of gliomas were identified as possible grade-specific indicators, and disrupted pathways give a thorough review of molecular signals implicated in glioma development [101].

Recently, to investigate the alterations in protein expression in GB, a meta-analysis of GB proteomics across 14 datasets was reported by Tribe et al.; in at least one of the datasets, the expression of 8801 proteins was shown to be elevated or reduced by at least 2-fold in GB samples compared to low-grade glioma or normal brain controls [102]. Figure 2 summarizes the principal findings.

### 3.7. Proteomics to Unravel the Resistance Mechanisms

As previously mentioned, MGMT confers resistance to TMZ therapy [58]. However, this resistance is inherent, and intriguingly, patients with acquired resistance to TMZ present with low levels of MGMT. Therefore, a study sought to examine the underlying mechanisms of the latter cause of resistance [103]. In an in vitro and in vivo mouse orthotopic brain tumor model, the role of the Down syndrome crucial region protein 3 (DSCR3) on MGMT-deficient GB was examined. Potential protein partners for DSCR3 were found using membrane-cytoplasmic separation of plasma membrane proteins and subsequent label-free quantitative proteomics. The reverse transport of solute carrier family 38 member 1 (SLC38A1) mediated by DSCR3 was demonstrated using immunofluorescence.

During TMZ therapy, DSCR3 was elevated in MGMT-deficient GB cells. Both DSCR3 and SLC38A1 were found to be substantially elevated in patients with recurrent GB. In MGMT-deficient GB cells, silencing DSCR3 or SLC38A1 expression can improve TMZ sensitivity. Proteomics and in vitro investigations demonstrated that the abundance of SLC38A1 on the plasma membrane is maintained through DSCR3-dependent recycling, resulting in tumor growth and acquired TMZ tolerance in MGMT-deficient GB [103].

### 3.8. Proteomics and GB Characterization

In the hope of unraveling the underlying mechanisms of GB, researchers are continuously working to break new ground. Rather than discarding the irrigation fluid used alongside brain tumor removal [104], researchers in Italy thought of preserving this fluid that is gathered in the cavitating ultrasound aspirator (CUSA) [105], as it comprises a valuable mix of tumor tissue fragments [104]. Samples from seven operative patients with IDH1 wild-type GB were obtained. The resourcefulness of this method was accompanied with proteomics analysis by LC–MS following the shotgun proteomic approach. The aim was to comparatively characterize the protein profile of CUSA fluid of recurrent and recently diagnosed GB from different areas of resection, i.e., tumor core and periphery. The identified proteins were then sorted to reveal elements not only unique, but also common to the tumor state or tumor zones. Gene ontology classification and pathway overrepresentation analysis were also performed. To reveal variation in expression, the proteins shared by the core and peritumoral areas were investigated further using relative quantitation. In comparison to the tumor periphery, nine proteins were identified that marked the core of newly diagnosed GB. These findings provide an overview of the proteomic characterization of CUSA fluid, and the profiles associated with the various sample groups [105]. As a result, CUSA fluid may be a novel source of potential biomarkers.

Similarly, quantitative proteomic mappings of primary and recurrent GB were investigated using SWATH-MS [106]. Immunohistochemistry was used to validate recurrence-associated proteins, which were then investigated in human glioma cell lines, human brain slice cultures, and orthotopic xenograft mouse models. Although overall proteomic changes were variable between patients, at recurrence, researchers noted BCAS1, FBXO2, and INF2 as consistently upregulated proteins and validated these with immunohistochemistry. In orthotopic xenograft mouse models, FBXO2 knockout imparted a significant survival benefit and reduced invasive growth in brain slice cultures [106].

In a similar effort, the subventricular zone GB cells (SVZ-GB) were investigated with cells on the initial tumor mass (TM) to identify specifically expressed proteins using MS, upon which B7-H3 was identified [107]. B7-H3 is a type I transmembrane protein [108]; known to be largely overexpressed in several cancer cells [109]. Human GB, non-malignant brain tissue, newly diagnosed and recurrent GB samples were used for the comparative analysis in the study. B7-H3 has two isoforms, 2IgB7-H3 and 4IgB7-H3 [108]. Although the former is exclusive in non-malignant brain tissue and the latter in GB, the expression of 2IgB7-H3 was noted to be greater in GB recurrences, as well as depicted increased TMZ therapy resistance. Therefore, 4IgB7-H3 is an intriguing contender for GB-targeted therapeutics, whereas 2IgB7-H3 may be linked with recurrence via chemotherapy resistance.

### 3.9. Proteomics for Identifying Biomarkers

An intriguing biological entity secreted by all cell types that assist in intercellular communication are the extracellular vesicles (EVs) [110]. These spheroid structures vary in their origin, bioactive content, and size, and are ideal for identifying disease markers due to being stable carriers of biological content such as proteins, DNA and mRNAs [111,112]. Therefore, extracellular vesicles (EV) have been of interest in terms of being used as a screening platform to identify markers related to early diagnosis, prognosis, and therapeutic response [113,114,115]. In a recent study, proteomic profiles of EVs sourced from human primary astrocytes (HPAs) in four different GB cell lines were analyzed through mass spectrometry. Results indicated elevated levels of sushi-repeat-containing protein X-linked (SPRX) in most GB-derived EVs, which was lacking in the EVs from HPA. The study was further able to deduce that the expression of SRPX is related to tumor grade, the expression of the SPRX gene was enhanced when GB cells were exposed to TMZ, and that the expression of the gene’s knockdown inhibits cell viability. Since multiple studies have established the role of SPRX in senescence [116], apoptosis [117], and cell migration and invasion [118], the study ultimately provided the possibility of using SPRX as a therapeutic target for GB, as well as a tumor marker for prognostic and diagnostic objectives [119]. 

Recently, Dahlberg et al. conducted a study to identify whether the microenvironment of cystic GB harbored growth-stimulating factors [120]. Cystic fluids from a total of 37 GB patients were obtained, where samples from 25 patients underwent hormone analysis and 12 underwent proteomics analysis. Interestingly, the GB cyst fluid representing the said microenvironment contains a slew of tumor growth factors, both non-hormonal and hormonal, varying in concentrations between patients when quantified. Growth hormone and testosterone concentrations were linked with tumor size, however, erythropoietin concentration correlated negatively with patient survival. Proteomics analysis was able to reveal between 314 to 901 different proteins, where 146 proteins were ubiquitous in all the 12 GB samples, and 70% were found to be plasma proteins, hinting at the disruption of the BBB in GB. The identified proteins ranged from those that participate in tumor growth and vascularization, possessing anti-apoptotic effects, to those that facilitate adaptation to hypoxia and participate in the inflammatory processes. However, proteomics did not give absolute concentrations of the proteins in the cyst fluid of GB. 

In another study, proteomic analysis helped unveil the underlying mechanisms that prompt the differentiation effect of CP-673451-an RTK inhibitor-on GB cells, particularly via the upregulation of dual-specificity phosphatase 1 (DUSP1) and downregulation of phosphorylated p38 mitogen-activated protein kinase (p38MAPK) [121]. Expression of the former has been linked to synaptic activity and is essential for dendritic development and axonal arborization, while the latter is one of the downstream targets of DUSP1, the activation of which inhibits neurite growth. CP-673451 is an inhibitor of PDGF, and could potentially be a differentiation agent useful in differentiation therapy for GB [122]. In hopes of counteracting the limited scope of current GB therapies, differentiation therapy aims to suppress tumors by converting undifferentiated cancer cells with high malignancy into differentiated cells with reduced malignancy [123]. The findings from the study imply that CP-673451 therapy may hold tremendous potential as part of a unique GB therapeutic strategy.

Likewise, a plausible therapeutic target and prognostic biomarker in GB was discovered. By utilizing the proteomics approach of tandem mass tag and TMT, and performing the ingenuity pathway analysis (IPA), the expressions of protein phosphatase 1γ (PP1γ), SRY (sex determining region Y)-box 2 (SOX2) and yes-associated protein 1 (YAP1) were identified from samples obtained from five GB patients. The proteins identified are considered to be the vital components of the Hippo signaling pathway in GB. This signaling system regulates tissue development and organ size [124]. Its principal effector is YAP1, that helps malignancies retain stem cell features and chemotherapeutic treatment resistance [125], and is a key driver of malignancy [126,127]. Being an important regulator of YAP1 dephosphorylation and thus having the propensity to influence oncogenesis [128], PP1γ was further investigated. It was concluded that PP1γ is a hub protein that is linked with the Hippo signaling pathway. Hub proteins have multiple binding sites and represent those nodes in a network that interact with several other nodes. In other words, hub proteins are central entities in a protein-protein interaction network [129]. Therefore, PP1γ could be an explicit therapeutic target for GB.

### 3.10. Pharmacoproteomics

Although immunotherapy is an effective approach to treating different malignancies, the highly immunosuppressive microenvironment that encompasses GB promotes tumor growth and renders immunotherapy ineffective against it [130]. However, oncolytic viruses (OVs) activate and increase host immune responses against tumor cells, eliminating remnant malignant cells and encouraging the formation of long-term antitumor immunity [131]. Long-term survival with OV therapy, however, is rare [132]. Therefore, researchers proposed that the identification of the secreted immunogenic proteins in response to GB samples treated with herpes simplex virus type 1 (HSV1), ex vivo, as a clinical model for an OV infection, would proffer a tool to predict sensitivity to the therapy [133]. Indeed, when the secreted proteome was analyzed by MS and the transcriptome by gene-microarray, an increase in the transcripts encoding for the secreted proteins were observed, suggesting a unifying theme in cancer cells’ response to infection. Moreover, the secretome proteins were found to be associated with B-cell-dependent immune response memory and T-cell-mediated cytotoxicity. Furthermore, the study provided thorough insight into the intricacy of GB immune response [133] and underscored the field of pharmacoproteomics; a rapidly expanding discipline analogous to pharmacometabolomics and in which proteomics methods are used to aid in the development of personalized medicine [134]. Further proteomic studies are summarized in Table 3.

## 4. Integrating Proteomics and Metabolomics

As previously mentioned, metabolomics and proteomics are not mutually exclusive, i.e., by adopting an integrative omics approach, different omic techniques can be utilized to bring forth “integrative information”. Moreover, integrated findings yielded from metabolomics and proteomics can identify key networks and signaling pathways that may be crucial in the metabolic regulation of certain biological proteins and hence can aid in identifying potential therapeutic targets. In this context, a study laid forth the possible mechanism of the invasive and migratory potential of GB [147].

As a tumor progresses, it expands in size, and the distance of oxygen diffusion in the blood vessels eventually increases, ultimately inducing hypoxia [148,149]. Multiple studies have confirmed that, among other factors, the hypoxic microenvironment surrounding tumor cells determines the invasiveness and migratory potential of GB, as well as its radiotherapy resistance [14,150,151,152,153,154,155].

In an effort to uncover the mediators behind this invasive and migratory property, metabolic and proteomic profiling on GB cell lines under hypoxic experimental conditions was carried out [147]. By conducting LC–MS/MS, researchers revealed that the metabolome of LN18 cells undergoes notable modifications under hypoxia as compared to normoxia [147]. In short, researchers discovered 1160 and 1958 significant differential anions and cations, which are primarily abundant in metabolic processes including the tricarboxylic acid cycle, glycolysis, lipid metabolism, amino acid, and nucleic acid metabolic pathway. Proteomic profiling further revealed that 62 out of the 2348 quantifiable proteins were significantly modified. The study also proposed a model of how P4HA1, an isozyme of 4-hydroxy-prolyl hydroxylase (P4H), triggers GB invasion in hypoxia. Figure 3 summarizes the proposed mechanism. Briefly, HIF1α is more abundantly expressed under hypoxic conditions. As the name suggests, Hypoxia-inducible factor 1-alpha (HIF1α) is a crucial transcriptional regulator of hypoxia signaling [156]. The upregulation of HIF1α expression further enhances the expression of nNOS (neuronal nitric oxide synthase), which in turn activates the expression of P4HA1, ultimately reprogramming epithelial–mesenchymal transition (EMT), and thus, increasing the invasiveness of GB [147].

## 5. Limitations

With the rapid advancement and implementation of high throughput technologies and bioinformatics, omics are being increasingly acknowledged by researchers. Despite the applications presented in this review, omics harbors a few limitations. Firstly, omics is untenable when it comes to its holistic utilization due to its implicit difficulty in interpreting, comparing, and combining data obtained from different settings. Moreover, the harmonization of methodologies and protocols is still insufficiently adopted across proteomics and metabolomics studies. Furthermore, given the absence of reference values for metabolites and proteins discovered by omics, biomarker discovery with an omics method is dependent on comparing patients to healthy controls, which is frequently opted in clinical trials (Table 4). This is a caveat, since, given the many physiological and environmental variables and interethnic differences [157], selecting a precisely matched control group of healthy participants can be challenging.

Moreover, the lack of gender and chronological age-specific reference values for different specimens [158,159] causes results to be expressed as values upregulated and downregulated in comparison to controls, making comparisons between studies more difficult. Experimental modeling of diffuse gliomas is challenging as well because well-established preclinical models, such as in vitro cell lines and xenografts, do not dependably mimic several features of glioma physiology [160].

## 6. Conclusions and Future Perspectives

A variety of techniques and analytical platforms are being used in metabolomics and proteomics to bring forth new insights into GB therapeutics. By summarizing recent studies utilizing metabolomics and proteomics in GB research in both preclinical and clinical settings, we aimed to emphasize the significant contribution of this technology in the realm of GB research and its potential to pave the way for accurate clinical prognosis, diagnosis, and therapy evaluation in the near future. 

Although metabolomics and proteomics are consonant with other omics techniques, profiling the metabolome can give insight into the impact of environmental conditions on cellular functions. Because the metabolome more accurately represents a cell’s actions at the functional level, it may be considered complementary to genomes, transcriptomics, and proteomics. Nevertheless, current efforts in biomarker discovery frequently focus on the identification of differentially expressed proteins and their link to a certain illness. Protein biomarkers’ actual mechanism of differential expression and functional relevance in GB are frequently unknown and understudied. 

Supplementing the platform with genomic, transcriptomic, and metabolomic data for integrated biomarker discovery can make use of data collected by the omics technologies and can also help plan the processes along the road to introduce regulatory-approved clinical assays and establish robust clinical guidelines for GB therapeutics. Therefore, arming an interdisciplinary team with certifications for data analysis and interpretation will speed the process of generating clinical applications that engender from the theoretical findings of metabolomics and proteomics, and lay the foundations for the ubiquitous use of integrative omics in research in the not-so-distant future.

## Figures and Tables

**Figure 1 ijms-24-00348-f001:**
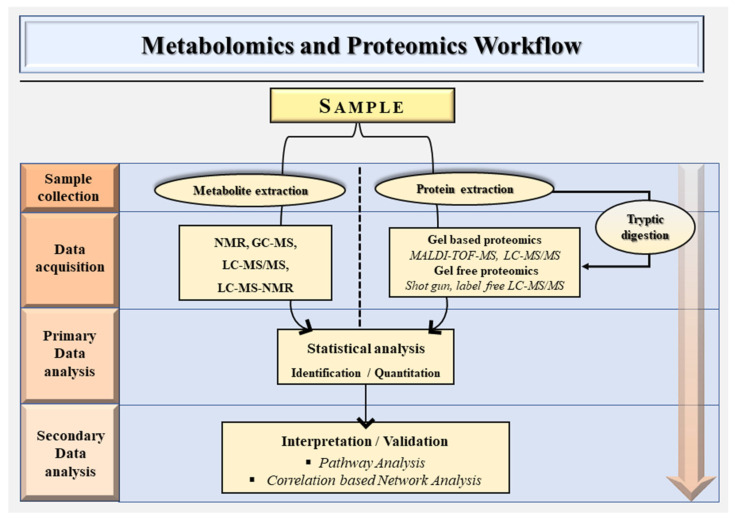
A general schematic overview of the metabolomics and proteomics workflow.

**Figure 2 ijms-24-00348-f002:**
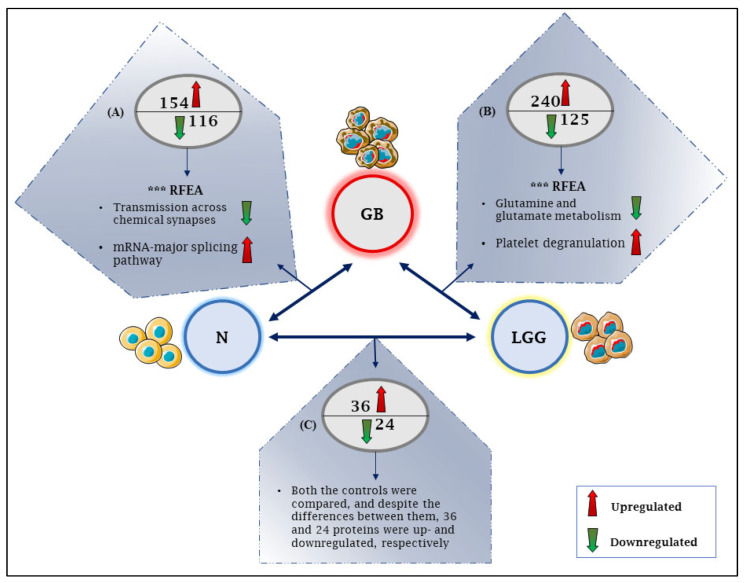
A schematic representation of differentially expressed proteins in GB as concluded from a meta-analysis across 14 datasets by Tribe et al. GB = glioblastoma, N = normal brain cells, LGG = low-grade gliomas, *** RFEA = most significant parent terms in the reactome functional enrichment analysis. (**A**) In two or more of the four datasets comparing GB to the normal brain, 154 proteins were upregulated, RFEA of which showed synaptic signaling to be the most significantly over-represented, and 116 proteins were downregulated with mRNA metabolism (mRNA splicing) to be the most significantly over-represented. (**B**) When comparing GB to LGG, 240 proteins were upregulated, RFEA of which showed platelet degranulation to be overrepresented, and 125 were downregulated in three or more of the ten datasets, with the significant parent term in RFEA involving glutamine and glutamate metabolism. (**C**) Between LGG and the normal brain controls, 36 proteins were elevated, and 24 proteins were downregulated.

**Figure 3 ijms-24-00348-f003:**
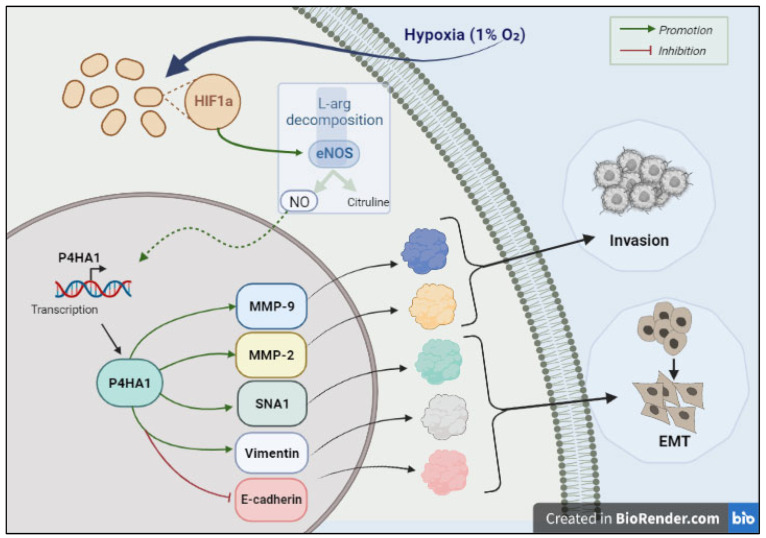
A diagrammatic illustration of the downstream signaling cascade associated with glioblastoma cell invasion induced by hypoxia. Created with BioRender.com.

**Table 2 ijms-24-00348-t002:** Summary of studies employing metabolomics in GB research.

**Application**	**Aim**	Tissue Type /Cell Line	Technique	Key Metabolites	Key Metabolic Pathways	Principle Insights	Reference
Biochemical characterization	Differentiate glioblastoma subtypes; define infiltrative tumor boundaries; potential utility in evaluating treatment effects.	Tumor and peritumoral GB tissue	MALDI-TOF-MSI (matrix-assisted laser desorption/ionization-time-of-flight-mass spectrometry imaging)	Antioxidants Fatty acids Purine and pyrimidine metabolites Reduced N-acetylaspartate abundance, etc.	Purine and pyrimidine metabolism, arachidonic acid synthesis, TCA cycle.	Metabolic information obtained could enhance and customize therapy methods The study underlines MSI’s appropriateness for GB research	[86]
Biochemical characterization	Create a xenograft for GB therapeutic testing; investigate the link between treatment efficacy and tumor metabolism.	Glioblastoma xenograft tissue	MALDI –Fourier transform ion cyclotron resonance (FT-ICR)-MSI	Heme ATP Acylcarnitine	Glycolysis, fatty acid metabolism, antioxidant, and anti-apoptotic functions.	Cells in the tumor’s core and edge experience distinct fatty acid metabolism, leading to different chemical microenvironments within the tumor.This can impact medication distribution via changes in tissue drug affinity or transport and is an essential consideration for therapeutic options in the treatment of GB.	[87]
Pharmacometabolomic approach	To investigate, for the first time, the influence of glabrescione B (GlaB), a known Hedgehog (Hh) pathway inhibitor, on glioma cell proliferation and metabolism in in vivo and in vitro models.	Murine glioma cells (GL261)	1H-NMR, HPLC–MS	Lactate Glycine Tyrosine Phenylalanine Histidine Alanine Leucine Isoleucine Valine	Glycolytic metabolism.	The endo- and exo-metabolomes of GlaB-treated and untreated cells exhibited changes in metabolite levels over time. GlaB, a direct inhibitor of the transcription factor Gli1, suppresses glioma cell proliferation while exacerbating the Warburg effect.	[88]
Comparative biomarker discovery	Because altered tumor metabolism is one of the hallmarks of cancer, the aim was to explore if the rate-limiting enzyme argininosuccinate synthetase (ASS1) positive and negative GB cell lines had distinct metabolic profiles that may allow for non-invasive diagnosis and reveal new treatment prospects.	GAMG LN229 SNB19 T98G U118 U87 Normal Human Astrocytes (NHA)	One and two-dimensional gas chromatography-time-of-flight mass spectrometry (1D/2D GC-TOFMS), LC-TOFMS.	Mannose Galactose Glucose Pyruvic acid Citrate α-ketoglutaric acid	Not detected.	The metabolome contains systematic information distinguishing between ASS1 positive and negative GB cell lines. There is a possibility of identifying metabolite biomarkers for the non-invasive detection from these subtypes, as well as the identification of novel treatment targets.	[89]
Pharmacometabolomic approach	The goal of this trial was to see if carboplatin worked in tandem with the mTOR complex 1 inhibitor (everolimus) in pediatric low-grade glioma (pLGG).	pLGG cell line BT66 JHH-NF1-PA1 Res259 Res18	LC–MS	Glutathione Glutamine Glutamate	Comparable pathways were discovered in patient-derived xenograft in mice.	The combination of everolimus and carboplatin works synergistically in pLGG. The study confirms a novel therapy regimen that may be promptly pushed into pediatric phase I/II clinical trials. This work presents a justification for novel mTORC1-based inhibitor therapy combinations in brain malignancies.	[90]
Pharmacometabolomic approach	To investigate the effect of glutaminase (GLS) inhibition on GSCs, which have been implicated in the development of medication resistance and tumor recurrence.	^1^H-NMR JHH520 GBM1 268, 407, 23, 233, 349 SF188 NCH644	^1^H-NMR	Alanine Aspartate Glutamine Glutamate Glycine Glutathione Lactate Myo-inositol Succinate Tricarboxylic acid Total choline	Not detected	The findings demonstrate the use of in vitro pharmaco-metabolomics for therapeutic effectiveness evaluation and compound risk assessment. It emphasizes the importance of GLS as a druggable and prospective therapeutic target in our desire to enhance the management of GB medication resistance and tumor relapse by focusing on GSCs subpopulation.	[91]
Developmental therapeutics	To develop standardized pediatric high-grade gliomas (pHGGs) models for drug testing and to generate an exact physiological brain environment in vitro.	Primary glioblastoma	NMR	Acetate Alanine Beta-glucose Choline Creatine Glutamate Glycerophosphocholine Glycine Lactate Myo-inositol N Acetylaspartate, Serine Taurine Valine	Some pathways were altered in the 2D/3D cell cultures pathways in patient tumor relapse.	A hypoxic environment helps to preserve the original patient tumor metabolism and characteristics. The multi-step effort may be regarded as a standard for developing therapeutically relevant models.	[92]
Developmental therapeutics	Researchers hypothesized that the branched-chain α-ketoacids (BCKA) depletion is caused by the (R) enantiomer of 2-hydroxyglutarate((R)-2HG)’s direct, competitive suppression of branched chain amino acids transaminases (BCAT) activity.	GSC lines: TS603, TS516, MGG152, TS676, BT054 BT260NHA HT1080 HOG IDH1 R132H mutant IDH2 R172K mutant HCT116 HEK293T NCI-H82	GC-MS, hybrid triple quadrupole mass spectrometer, Hydrophilic interaction liquid chromatography(HILIC)	Alpha-Keto-beta-methylvalerate Alpha-Ketoisocaproate Glutamate 2-hydroxyglutarate Leucine Valine isoleucine	Increased BCAT activity in vitro and in vivo.	BCAT suppression produces metabolic vulnerabilities that can be leveraged therapeutically to sensitize IDH mutant gliomas. ((R)-2HG is overproduced in IDH mutant GBs). Gliomas with IDH mutations are more sensitive to radiation when combined with glutaminase inhibition, suggesting a novel way to treating these tumors.	[93]
Developmental therapeutics	To assess the effect of paclitaxel and/or etoposide on the molecular changes in GB cells	U87 U373	Ultra-high-performance liquid chromatography-electrospray ionization quadrupole time-of-flight mass spectrometry (UHPLC-ESI-QTOF-MS)	Nutriacholic acid L-phenylalanine L-arginine Guanosine ADP Hypoxanthine guanine	Urea and citric acid cycles Metabolism of polyamines and amino acids	The results can be used to map the anticancer activity of paclitaxel and/or etoposide within the cancer cells under investigation.	[94]
Biomarker discovery	To use NMR spectroscopy to characterize the metabolome of tiny EVs or exosomes produced from distinct GB cells and compare them to the metabolic profile of their parental cells.	NHA U118 LN-18 A172	^1^H-NMR	Asparagine Acetone Carnitine Ethanol Formate Glycerol malate GSSG GSH GABA G6P Glucose Isoleucine Taurocholic acid Niacinamide lactate 5-oxoproline Citrate Proline succinate Homoserine Glycine	Not detected	The findings revealed a distinct divergence in the metabolic profiles of GB cells, EVs, and medium. The findings are reviewed in relation to new GB diagnostics and therapy monitoring.	[95]
Biomarker discovery	To describe a transcriptional adaption regulatory system that is influenced by environmental factors.	Primary GB	1H-NMR	Alpha-ketoglutarate Arginine Caproic acid Choline Dodecanoic acid Fructose Fumarate Glyceraldehyde Glutathione Glycine Guaiacol Glucose-6-phosphateLysine Succinic acid Serine Selenomethionine	Comparable metabolic environment spatial disparities	A multi-regional examination of a glioblastoma patient biopsy indicated complex metabolic landscape with varied degrees of hypoxia and creatine enrichment. In creatine-enriched settings, the glycine cleavage system, and hypoxia-inducible factor-1α (HIF1A) destabilization were changed, resulting in transcriptional adaptability.	[96]
Biomarker discovery	To test the hypothesis that GB plasma metabolite profiles may predict clinical outcomes.	Primary and recurrent glioblastoma	LC- triple quadrupole- MS	Arginine Kynurenate Methionine	N/A	The study discovered numerous plasma metabolites that are predictive in glioblastoma patients.	[97]
Biomarker discovery	To investigate the effects of a survivin inhibitor (pro-apoptotic effect) on the metabolome of primary GSCs to look for treatment response signals.	GSCs cultures established from IDH-wildtype GB tumor	NMR spectroscopy	Citrate Lactate	N/A	In comparison to spectrometry-based proteomics, the metabolomics technique used generated alternative biomarker possibilities, highlighting the benefits of complementary approaches. Citrate and lactate are magnetic resonance spectroscopy (MRS) -visible, therefore, these first findings provide the groundwork for further research into in vivo MRS of brain malignancies. NMR metabolomics, when combined, is a technique for tackling glioblastoma.	[98]

**Table 3 ijms-24-00348-t003:** Summary of studies employing proteomics in GB research.

Application	Aim	Tissue Type /Cell Line	Technique	Key Proteins	Key Pathways	Principle Insights	Reference
Pharmacoproteomics	To understand the extensive regulation of glioma metabolism in response to Delta-24-RGD (an E1A mutant oncolytic adenovirus) infection, by performing a cell-wide study of cytosolic, nuclear, and secreted proteomes during the early time course of the infection.	U87	Using iTRAQ, a shotgun comparative proteomic analysis of cytosolic fractions	Cytosolic proteins: Serine hydroxymethyltransferase, mitochondrial lactotransferrin Alpha-2-macroglobulin Proliferating cell nuclear antigen etc. Nuclear proteins: Programmed cell death protein 6 Replication factor C subunit 2 Annexin A2 Ribosome-binding protein 1 etc.	Proteostasis pathway Protein kinase C, ERK1/2, and p38 MAPK pathways	The findings assist in understanding the methods through which Delta-24-RGD exploits glioma proteome organization. Further exploration of this proteomic resource may lead to the development of complementary adenoviral-based vectors with increased specificity and efficacy against glioma.	[135]
Unraveling GB pathophysiology	To define the role of cat eye syndrome critical region protein 1 (CECR1) in tumor associated macrophages (TAMs) through a proteomic investigation of siRNA-mediated CECR1 silencing in THP-1-derived macrophages co-cultured with or without glial tumor cells.	U87 THP-1 cells (human monocytic cell line)	Mass spectrometry	ISG15 HLA-A HLA-B HLA-C TAP1 TAP2 TAPBP TIMP-1 WDFY1 SEPT7 S100A9 PLAU LAT2	MHC I antigen presenting pathway, Phagosome maturation, caveolin-mediated endocytosis, and type I interferon signaling pathways	CECR1-mediated molecular pathways and essential molecules operate in macrophages and glial TAMs (Tyro3, Axl, Mer, family of receptor tyrosine kinases). The proteome dataset might be used to create novel therapeutic targets for future immunotherapy research in the treatment of malignant (glial) cancers and autoimmune illnesses.	[136]
Unraveling GB pathophysiology	The constitutively active epidermal growth factor receptor (EGFRvIII) is an oncogenic factor that fuels GB aggressiveness and is ascribable to the release of extracellular vesicles (EVs). Researchers aimed to examine the effect of this oncogene on the profile of glioma EVs.	U373 U373vIII	MS	CD44/BSG TSPAN8 CD151 CD81 CD9 SDCB1 Actin, GAPDH CD44 ITGA6, ITGB4 TGFB1 Laminins, collagens	Vesiculation pathways	CD44/BSG were co-localized in cellular filopodia and EVs generated by EGFRvIII-expressing cells were double positive for these proteins. Oncogenic EGFRvIII alters the proteome and uptake of EVs related to GB.	[137]
Developmental therapeutics	To perform a comparison of the ligandomes of GSC and patient samples with the goal of discovering GSC-associated targets that are also present on primary patient malignancies.	Primary GB Peripheral blood mononuclear cells (PBMC) GSC cell lines: GS-2 GS-5 GS-9	LC–MS/MS	Cancer testis antigen (CTA) SERPINE1 FABP7 PTGFRN ALFPERITV (ATAT1) RLAPFVYLL (HEPACAM) SILDIVTKV (RFTN2)	Not available	The study identified a novel panel of T cell antigens characterized by the exclusive identification of malignant specimens, a substantial prevalence of presentation, and presence on the GSC compartment by mapping the HLA peptidome of glioblastoma and GSC. Epitopes of functional CD8 T-cell responses were identified, making them excellent candidates for immunotherapy.	[138]
Biomarker discovery	To investigate, in a cellular rat model, the line GL261.	3T3-L1 adipocyte cell line GL-261 murine model	MALDI-TOF-MS	Arbonic anhydrase Aldose reductase Endocan HGF IGF-I IL-6 IL-11 LIF PAI-1 SerpinE1 TNF-α TIMP-1 VEGF	Not available	STI1, hnRNPs, and PGK1, overexpressed otherwise in cancer, were under expressed. Carbonic anhydrase and aldose reductase, both of which play significant roles in inflammation, and cancer metabolism, are also reduced in glioma cells cultured in an adipokine-enriched environment, displaying a paradoxical association of a protective function between fat and cancer.	[139]
Biomarker discovery	To utilize SWATH-MS and quantitative targeted absolute proteomics to find plasma biomarker candidates for GB patients (QTAP).	Cyst fluid samples of IDH wildtype GB Non-cancerous brain tissue samples	SWATH-MS LC–MS/MS	LRG1 C9 CRP SERPINA3 APOB GSN IGHA1 APOA4	Not detected	To evaluate the links between biomarker candidates and GB Biology, the study looked at associations between biomarker candidate plasma concentrations and clinical presentation (tumor size, overall survival time, etc.) in patients. LRG1, CRP, and C9 plasma concentrations all revealed strong positive relationships with tumor growth.	[140]
Biomarker discovery	To identify and describe the effective biomarkers present in the small extracellular vesicles (sEVs) to improve GB diagnosis, and ultimately, patient prognosis.	Blood samples from GB patients and controls	MS, MS/MS	VWF FCGBP C3 PROS1 SERPINA1	B-cell receptor signaling pathway Pathways involved in complement activation, innate immune response, and platelet degranulation	Overall, the development of a non-invasive liquid biopsy method for the discovery of valuable biomarkers that could considerably enhance GB diagnosis and, as a result, patients’ prognosis, and quality of life, is promoted through this study.	[141]
Developmental therapeutics	To study GB- associated surfaceome by comparing it to the surfaceome of astrocyte cell lines in order to find new GB-specific targets.	NCH82 U-87 MG	MALDI-mass spectrometry	Plaur B41 alpha chain (HLA-b) A-24 alpha chain (HLA-a) DP beta 1 chain (hla-dpb1). CADM3 CADM4 NRCAM	Cell contact, cell adhesion, vascularization, and proliferation pathways	11 distinct potential GB targets were discovered, including 5 altered proteins such as MHC I, CYBA, EGFR, and RELL1.	[142]
Optimize GB datasets	To boost the translational importance of the Q-Cell datasets and to create a platform for academics to conduct rigorous preclinical neuro-oncology research.	Primary GB characterized cell line (Q-Cell)	LC–MS	BAH1 GSN JK2 MMK1 MN1 NNMT PLP2 PRDX6 RN1 SB2b SB2 SOD2 SERPINE1 WK1	PI3K and mTOR signaling pathways TCA cycle, NFKB, and MAPK signaling.	In-depth proteomic characterization of the GB Q-Cell resource was obtained, serving as a dataset for future biological and preclinical research.	[143]
Establishing research methodology	To develop methods to analyze the proteome of small extracellular vesicles (sEVs) from low serum volume that is obtained from mice, to perform a longitudinal analysis of disease models.	Adult C57BL/6J mice murine glioma GL261 cell line	LC–MS/MS	Tetraspanins integrins Sdcbp Hspa8 Cd9 Itga2 Anxa4, Anxa5, Anxa7 Vamp8 Lrp1 Cpn1 Mhy9 Tln1 Tfrc, CD71 Apoc4	PI3K/AKT pathway	The methodology allowed for the identification and quantification of 274 protein groups. The longitudinal study discovered 25 altered proteins in GB serum sEVs, including proteins previously linked to GB development and metastasis.	[144]
Identification of resistance mechanism	To investigate the cytoplasmic proteome of U87 GB cells treated with TMZ, using bioinformatic approaches to thoroughly evaluate the raw data.	U87	Liquid chromatography–electrospray ionization–tandem mass spectrometry (LC–ESI–MS/MS)	DHX9 HNRNPR RPL3 HNRNPA3 SF1 DDX5 EIF5B BTF RPL8	Thyroid hormone, p53 and the PI3K-Akt signaling pathways Regulation of actin cytoskeleton	Dysregulation of spliceosome-related proteins SF-1, DDX5, and HNRNPR may all contribute to a disruption in DHX9 synthesis, eventually leading to GB TMZ resistance.	[145]
Morphoproteomics	To create a spatially conserved proteomic atlas of GB by meticulous microdissection and LC–MS/MS profiling of the traditional histomorphologic characteristics of the malignancy.	MYC-enriched cell lines (3-CI-AHPC, CD-437)	LC–MS/MS	Immunoglobulin CD276 (B7-H3) AKAP12 PTPRZ1	Hypoxia pathway axis	Various glioblastoma locations may be divided into subpopulations in which glioma cells favor migration and infiltration above proliferation and growth.	[146]

**Table 4 ijms-24-00348-t004:** Clinical trials employing omics technology on different diseases (adopted from clincaltrials.gov, accessed on 3 September 2022).

Disease	Intervention/Treatment	Country	ID	Status	Outcome to Assess (in Relation to Metabolomics and Proteomics)
Prostate cancer	N/A	Taiwan	NCT03237026	Recruiting (as of September 2022).	Biochemical recurrence or progression.
Hepatocellular Carcinoma	Procedure: surgical resection drug: adjuvant atezolizumab–bevacizumab Therapy	Singapore	NCT05516628	Recruiting (as of September 2022).	Biomarkers based on multi-omics (epigenomics, genomics, transcriptomics, immunomics, proteomics, and metabolomics) and spatial tumor microenvironment profiles of both tissue and peripheral blood that predict therapy response.
Urothelial Carcinoma	Biomarkers and proteomics	Italy	NCT04770974	Not yet recruiting (as of September 2022).	Bladder cancer’s metabolomic profile.
Pancreatic Neoplasms	Diagnostic test: soluble biomarkers dosage	France	NCT04370574	Recruiting (as of September 2022).	To identify biomarkers that seem to be prognostically significant on overall survival or disease independent in pancreatic cancer patients.
Metastatic Urothelial Carcinoma	N/A	Taiwan	NCT04641936	Recruiting (as of September 2022).	To discover possible metabolite and protein indicators capable of predicting the success and side effects of immuno–oncology-based therapies.
Adrenal Neoplasm Endocrine Tumors Neuroblastoma Parathyroid Neoplasms Thyroid Neoplasms	N/A	Unites States	NCT01005654	Recruiting (as of September 2022).	To create a metabolomic, proteomic, genetic, and epigenetic profile of endocrine neoplasm that would allow discriminating between benign and malignant tumors in each of the endocrine histologies under investigation.
Metastatic Renal Cell Carcinoma	N/A	Taiwan	NCT04712305	Recruiting (as of September 2022).	To discover possible metabolite and protein indicators capable of predicting the success and side effects of immuno–oncology-based therapies.
Significant Prostate Cancer	Dietary supplement: multi-carotenoids (MCS)	Taiwan	NCT03237702	Recruiting (as of September 2022).	Evaluating the effect of urine omics tests (metabolomics and proteomics) in participants undergoing or having undergone prostate biopsy and/or subsequent MCS supplementing
Colorectal Cancer (CRC)	N/A	United States	NCT00898378	Completed	Utilize biological samples from patients with CRC or colorectal adenomatous polyps, as well as those without polyps, to perform genomic, metabolomic, lipidomic, glycoproteomic, and proteomic profiling to create an omic profile.
Sarcoma Endocrine Tumors Neuroblastoma Retinoblastoma Renal Cancer	N/A	United States	NCT01109394	Recruiting (as of September 2022).	To conduct on tumor and normal tissues, systematic molecular, genomic, proteomic, metabolomic, and other high throughput (Omics) profiling.
Extrapulmonary Small Cell Cancer Non-Small Cell Lung Cancer Small Cell Lung Cancer Pulmonary Neuroendocrine Tumors Thymic Epithelial Tumors	N/A	United States	NCT02146170	Recruiting (as of September 2022).	Conduct genomic, proteomic, and immunological investigations on blood, tumor, bodily fluid, and normal tissue to identify new therapeutic drugs, innovative treatment techniques, and new prognostic and diagnostic markers.
Thyroid Nodule Thyroid Cancer	Diagnostic test: multi-omic analyses of blood and surgical specimens	Italy	NCT05428371	Not yet recruiting (as of September 2022).	Identification of biomarkers of thyroid carcinoma.
Cutaneous Squamous Cell Carcinoma Basal Cell Carcinomas	Biopsy	France	NCT04389112	Recruiting (as of September 2022).	Metabolic profiling of the carcinogenesis stages of glycolysis, oxidative phosphorylation.

## Data Availability

Not applicable.

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
