# Peer review of "Preclinical and Clinical Applications of Metabolomics and Proteomics in Glioblastoma Research"

_ijms, 2022, doi:10.3390/ijms24010348_

Round 1

Reviewer 1 Report

The authors systematically summarize the application of metabolomics and proteomics in the study of GB disease. GB is a very serious neurological pathology, and thus the content summarized in this manuscript has a broader reference value. In particular, the content summarized in Tables 3 and 4 is very informative, and the key metabolites and protein markers summarized therein are very meaningful. This information provides ample reference information for researchers who carry out the omics exploration of GB. Furthermore, the manuscript is well written and is worthy of publication. But the writing in some section is not clear enough.

Minor concerns:

1.       Fig1, the data acquisition methods were confusing. MS was never be used alone as a metabolite analysis method. GC-MS/MS is rare, but GC-MS is very common.

2.       Table 1, the row of “MS” is unnecessary, since it usually been used coupling with LC.

3.       Table 1, the row of "GC-MS/MS” should be GC-MS according to the content.

4.       Table 2 is unclear. ESI, MALDI are ion source types. SWATH is data acquisition method. iTRAP is a detector type. These cannot be mixed into a discussion of similar matters.

5.       Section3.4, There is no connection between the content of this part and the subtitle. The pharmacological research of the three papers is summarized here, which is far from clinical treatment. What does “developmental” mean and what does it have to do with the discussion in this part?

Reviewer 2 Report

The paper is a good summary of studies, including more recent ones, utilizing metabolomics and proteomics in GB research in both preclinical and clinical settings with the aim of underline a possible application of this technology for accurate clinical prognosis, diagnosis, and therapy evaluation in the future while also emphasizing the limitations.

Some suggestions are given below:

-        -  In the introduction when the incidence of glioblastoma is mentioned, reference number 2 is a good work but not very recent. Please add more recent paper if it is possible.

-         -  In paragraph 1.2 of the introduction it could also be underlined that in a subgroup of patients, the metabolic features of GBM cells, as assessed by NMR analysis, may be informative of the genomic/proteomic landscape of the tumor, and ultimately of the GBM subtype and clinical outcome  as demonstrated by Marziali, G., Signore, M., Buccarelli, M. et al. Metabolic/Proteomic Signature Defines Two Glioblastoma Subtypes With Different Clinical Outcome. Sci Rep 6, 21557 (2016). https://doi.org/10.1038/srep21557

-         -  Check the title of reference number 18
